# Comparative Study on Mechanical Properties and Microstructure Evolution of Mg-3Zn-1Mn/Sn Alloy through Ca-La-Ce Addition

**DOI:** 10.3390/ma17194840

**Published:** 2024-09-30

**Authors:** Ke Hu, Tingting Tian, Yunfeng She, Xiaoming Guo, Lixia She, Junjie Huang, Xiaomin Huo, Xiao Liu, Zhaoting Xiong, Chao Lu

**Affiliations:** 1School of Materials Science and Engineering, Guangdong Ocean University, Yangjiang Campus, Yangjiang 529500, China; hukee12345@126.com (K.H.); ttt0319@126.com (T.T.); 13612817508@163.com (X.G.); huangjunjie1027@126.com (J.H.); xiong_868@163.com (Z.X.); 2School of Chemical Engineering, China University of Mining and Technology, Xuzhou 221006, China; 3Changxing Yunfeng Luliao Co., Ltd., Huzhou 313000, China; rzms@163.com (Y.S.); wangxiaojiao2001@126.com (L.S.); 4Taizhou Vocational and Technical College, Taizhou 318000, China; huoxm0928@163.com; 5College of Marine Equipment and Mechanical Engineering, Jimei University, Xiamen 361021, China; liuxiao0105@163.com

**Keywords:** Mg-3Zn-Sn/Mn alloy, strain-hardening exponent, extrusion, texture evolution, Mg alloy

## Abstract

This study systematically investigates the influence of the composite addition of Ce, La, and Ca elements on the microstructure evolution and mechanical properties of Mg-3Zn-1Mn/Sn (wt.%) alloys. It indicates that the strength of Mg-Zn-Mn series alloys is superior to that of Mg-Zn-Sn series alloys, due to the stronger restriction of nanosized Mn particles on the recrystallization process and grain growth compared with Mg_2_Sn phases. The addition of the Ca-La-Ce elements significantly enhances the strength of the Mg-3Zn-1Sn alloy (YS increased by approximately 92.5%, UTS increased by approximately 29.2%, and EL decreased by nearly 52.2%), while for the Mg-3Zn-1Mn alloy, a balanced effect on both the strength and performance can be achieved. This difference mainly lies in the more pronounced refined effect on the grain size and the formation of a bimodal grain structure with strip-like un-DRXed grains and surrounding fine DRXed grains for the Mg-3Zn-1Sn alloy. In contrast, the addition of the Ca-La-Ce elements has a less obvious hindrance on the recrystallization process in the Mg-Zn-Mn series alloy, while significantly weakening the extrusion texture while refining the grains. Through in-depth characterization and experimental analysis, it is found that Sn and Ca can promote the formation of brittle and fine secondary phases. A nanoscale Sn phase (Mg_2_Sn phase) is more likely to accumulate at the grain boundaries, and the size of the nanoscale Ca_2_Mg_6_Zn_3_ in Mg-Zn-Mn series alloys is finer and more dispersed than that in Mg-Zn-Sn series alloys, thus strongly hindering recrystallization and refining the recrystallized structure of the alloy.

## 1. Introduction

As the lightest green metal material in the world, magnesium alloys with low density and high specific strength possess significant commercial value in fields such as aviation, automotive, and electronics [1,2]. However, the hexagonal close-packed (HCP) crystal structure limits the effective slip systems of magnesium alloys, leading to significant unisotropy among different crystal planes. Consequently, issues such as poor formability, large performance differences, and difficulty in balancing strength and ductility arise, severely restricting the long-term industrial application prospects of magnesium alloys. How to further improve the mechanical properties of magnesium alloys, simultaneously coordinate their strength and ductility, and enhance the corrosion resistance is a long-term research goal that urgently needs to be developed. It holds crucial significance in promoting their large-scale commercial application [3].

Currently, the method of manipulating alloy microstructures by incorporating trace alloying elements stands as an efficacious means to enhance the mechanical properties [4]. Among all additional elements, Mn, Sn, and Ca emerge as species capable of directly precipitating nanoscale particles during the hot deformation processes of magnesium alloys. The individual and synergistic additions of these elements markedly elevate the yield strength (YS) of the alloy [5,6,7,8,9,10,11,12,13,14]. For instance, Wang et al., scrutinized a low-Mg-1.2Zn-0.1Ca (wt.%) alloy and achieved an outstanding yield strength [5]. Li et al., elucidated the multifaceted role of nanoscale precipitates in conferring both an ultrafine grain size and ultrahigh strength to conventionally extruded Mg-Ca-Al-Zn-Mn alloys, yielding a tensile strength of 449 MPa, a yield strength of approximately 435 MPa, and an elongation of approximately 4.2% [11]. Gaowu Qin et al., found that the addition of Mn, Sn, and Ca simultaneously could increase the strength of the magnesium alloy to 450 MPa with 5% ductility through the mechanism of dispersion strengthening [7]. Xia et al., upon the foundation of elemental modification in the ZK61 alloy, concurrently introduced substantial quantities of Ca, Mn, and Sn elements, coupled with thermal extrusion processes and aging treatments, effectively refining the microstructure and engendering the precipitation strengthening of the β′ phase, resulting in a significant enhancement of approximately 36% in the yield strength of the ZMT614-0.5Ca alloy [12]. Subsequently, Xia Chen et al., further augmented the strength of the ZK61 alloy to 400 MPa by incorporating substantial quantities of Ca, Mn, and Sn elements [13,14].

Although the preceding research heralds the remarkable properties of magnesium alloys, it warrants attention that the sluggish deformation process significantly inflates production costs. Moreover, the extensive incorporation of Zn, Ca, Mn, and Sn elements exacerbates the alloy’s thermal contraction, unfavorably impacting the castability and weldability. Consequently, enhancing both the holistic performance and processability of magnesium alloys stands as a pivotal engineering pursuit. Composite alloying methodologies are instrumental in fostering a favorable comprehensive performance in magnesium alloys. Notably, for magnesium alloys inherently deficient in ductility, strength augmentation invariably entails concomitant reductions in processing efficiency and ductility. Recent investigations underscore the efficacy of rare-earth element additions in ameliorating the plasticity of low-zinc magnesium alloys. For instance, studies by B. Langelier et al. [15] elucidate the substantial grain refinement and concomitant strength and ductility enhancements achieved through Ca-Ce composite alloying in Mg-4Zn alloys. Similarly, Yuzhou Du et al., observe that varying doses of La microalloying refine the dynamic recrystallization grain sizes in Mg-Zn-La alloys, thereby enhancing ductility [16]. This strategic augmentation of strength at the expense of ductility delineates a realm of exploration in these alloys. Our research endeavors corroborate these findings, indicating that at an extrusion temperature of 300 °C, a Mg-2Zn-0.4Mn-1MM alloy manifests commendable comprehensive mechanical properties [17,18].

However, the predicament lies in the stringent processing conditions required for the performance of low-zinc alloys, and the deficiency of zinc diminishes the potential for further aging heat treatment of the alloy. Additionally, the extrusion force demanded by low-zinc high-strength alloys is substantial, which hampers production efficiency. These issues, though pragmatic, are frequently overlooked within the academic sphere. The Mg-3Zn series alloys represent a lesser-studied category within magnesium alloys, primarily because alloys within the kinds of alloys, such as Mg-3Zn and Mg-4Zn, exhibit moderate strength. However, Mg-3Zn stands out for its excellent welding properties and contains sufficient Zn content to facilitate subsequent heat treatment operations. Therefore, this study employs a Mg-3Zn alloy and employs a composite alloying approach to enhance its mechanical properties. Simultaneously, it investigates and compares the effects of La, Ce, and Ca combinations on the microstructural evolution and mechanical properties of Mg-3Zn-1Sn and Mg-3Zn-1Mn alloys. We hope that this study can provide data and theoretical support for the large-scale application of magnesium alloys in the future.

## 2. Experimental Procedure

The nominal and actual compositions of the four alloys studied in this work are shown in Table 1. The alloys were manufactured using a vacuum melting method. Industrial pure magnesium (99.9% wt.%), commercial pure Zn (99.9% wt.%), Mg-30LaMM alloy (wt.%, La:Ce = 2:1), Mg-10Mn (wt.%), and Mg-10Sn (atmosphere at 710 °C) were used to prepare the magnesium alloy materials required for this experiment. After stabilization, the molten alloy was poured into preheated low-carbon steel molds maintained at 150 ± 10 °C for 12 h to obtain homogeneous alloy ingots with a diameter of 100 mm and a height of 120 mm. Subsequently, the ingots underwent homogenization annealing treatment at 350 °C for 4 h. Using an LXJ-500T extrusion machine (Wuxi Dazhou Machinery Co., Ltd., Wuxi, China), the alloy was extruded into rods with a diameter of 12 mm under the following conditions: temperature of 300 °C and extrusion speed of 5 mm/s, with an extrusion ratio of 17.5:1. Using the extruded rods as raw materials, tensile specimens were prepared according to ASTM standard B557M-10 [19], with a gauge diameter of 6 mm and a gauge length of 25 mm. The tensile direction was aligned parallel to the extrusion direction. To ensure reproducibility, a minimum of three tests were conducted. Tensile testing was performed using a tensile test mechine (ZwickRoell Z100, Baden-Wurttemberg, German) at room temperature with a speed of 1 mm/min at room temperature (25 °C), following the procedures outlined in ASTM standard B557M-10.

For the hardness test, the micro Vickers hardness of the alloy was measured using a JMHVS-1000ZCCD hardness tester (Shanghai Precision Instrument Co., Ltd., Shanghai, China) with a 200 g load for 10 s (HV0.2). At least five points near the center of the sample were selected for averaging to reduce errors.

According to ASTM E3-11 [20], metallographic samples of the alloys were prepared. Specifically, the etchant solution was prepared using a 4% solution of picric acid, hydrochloric acid, and glacial acetic acid in ethanol (3.0 g picric acid, 2.5 mL glacial acetic acid, 5 mL water, and 50 mL ethanol). After grinding the samples with 800#, 1500#, 2000#, and 5000# sandpaper, mechanical polishing and etching were conducted on the castings, homogenized samples, and extruded specimens prior to metallographic examination. For characterization and analysis of the microstructure, the distribution of microstructures in the alloy after extrusion was observed using an optical metallographic microscope (OM, A LEICA DM 2500, Leica Camera AG, Wetzlar, Germany). Subsequently, the composition of phases in the alloy was analyzed using an X-ray diffractometer (XRD-6100, Shimadzu Corporation, Kyoto, Japan). The microstructure observations of the alloy after tensile testing were conducted using a scanning electron microscope equipped with an energy-dispersive X-ray spectrometer (SEM, Apreo 2 SEM, thermo Fisher Scientific Inc., Hillsboro, OR, USA). The average grain size of the samples was determined using the linear intercept method. To observe the texture distribution in the alloy, an electron backscatter diffraction spectrometer (EBSD, MAIA3 model 2016) was employed. The samples were cut into a thin slice with a thickness of 0.5 mm under the transmission electron microscope (TEM, FEI TECNAI G2 F20, thermo Fisher Scientific Inc., Hillsboro, OR, USA). It was ground and polished to a thickness of ~50 nm, and then perforated by argon ion milling.

## 3. Results and Discussion

### 3.1. Metallographic Examination and XRD Pattern Analysis

Figure 1 presents the optical microscopy (OM) images of the four samples. Overall, after extrusion, except for the ZT31 alloy, the other three alloys exhibit incomplete dynamic recrystallization (DRXed) grains. Among them, ZTXE3101 has the lowest degree of recrystallization, approximately 67.2%. It can be observed that the three samples with unrecrystallized (un-DRXed) grains have significantly different morphologies: the un-DRXed grains in the ZM31 alloy are coarse and elongated along the extrusion direction, while those in ZMTE3101 are also elongated but finer. In contrast, the un-DRXed grains in ZMXE3101 exhibit a blocky shape.

To elucidate the phase composition of the four alloy specimens, XRD analysis was conducted, and the results are depicted in Figure 1. Overall, the ZTXE3101 sample exhibits the most diverse precipitated phases, with a total of five phases present (Figure 1(d,d1)). For the ZM31 and ZT31 alloy samples, in addition to the common presence of the Mg matrix and MgZn_2_ phases, there are also precipitations of Mn and Mg_2_Sn phases, respectively. Compared to the alloys without rare-earth additions, the ZTXE3101 and ZMXE3101 alloys show the disappearance of the MgZn_2_ phase, replaced by Ca_2_Mg_6_Zn_3_ and RE phases (La, Ce)(Zn, Mg)_12_.

### 3.2. Microstructure Analysis by SEM and TEM

Figure 2 shows the SEM images and EDS analysis results of the typical second phases in the four alloys. From the figure, it can be observed that the second phases are crushed during the extrusion process and distributed along the extrusion direction. Among them, the ZM31 alloy has the smallest amount of second phases, with only a few irregularly shaped second phases sporadically distributed in the matrix. The EDS analysis reveals that these phases are Mg_2_Zn phases as shown in Table 2.

The number of second phases in ZTXE3101 and ZMXE3101 are significantly higher than those in ZM31 and ZT31. Although ZT31 appears to have a low number of second phases at low magnification in the SEM, the XRD and EDS analyses reveal that it contains three typical second phases: spiral Mg-Zn phases, punctate Mg-Sn phases, and fragmented Mg-Zn-Sn phases.

For the ZTXE3101 and ZMXE3101 alloys, the coarse phases are Mg-Zn-RE phases. A valuable finding is that when Ca is added to the Mg-Zn-RE phases, forming a Mg-Zn-RE-Ca phase, it becomes more easily fragmented during extrusion, resulting in elongated, band-like structures distributed along the extrusion direction.

Based on the metallographic graph, most samples exhibit partially recrystallized structures. The transmission electron microscopy (TEM) image clearly shows that during the extrusion process, dense Mn particles are formed in the Mn-added samples. These Mn particles, smaller than 1 μm, are uniformly distributed on the Mg matrix, restricting dislocation movement and hindering recrystallization and grain growth. For the ZM31 samples, in addition to Mn particles, nanosized Mg_2_Zn phases are also present, as shown in the high-resolution image (Figure 3c). These phases initially form as short rods and grow faster along the long axis during the growth process, resulting in a gradual increase in the aspect ratio.

In the microstructure of the ZMXE3101 alloy, in addition to nanoscale Mn elements, the presence of Ca_2_Mg_6_Zn_3_ phases with dimensions ranging from 10 to 100 nm has been observed. According to the previous literature, this phase exhibits strong pinning effects on dislocations and grain boundaries, inhibiting the recrystallization and subsequent growth processes of the material [12,13,21,22]. In the ZT31 alloy, a small amount of a Mg_2_Sn phase is formed, which has a much larger volume than the Mn particles (Figure 3g). Therefore, its quantity is relatively small, and its hindrance to recrystallization is minor. Whether through metallographic or EBSD analysis (refer to Section 3.4), the average grain size of the ZT31 alloy is higher than that of the ZM31 alloy. Chaoyue Zhao et al., believed that the role of Sn in Mg to generate nanoscale Mg_2_Sn for grain refinement is mainly to promote recrystallization nucleation [23]. Figure 3h–l are the TEM images of ZTXE3101, from which it can be seen that the nanophases of this alloy are mainly the Ca_3_Mg_6_Zn_3_ phases that are much more fine and dense than those in ZMXE3101. A large amount of literature has reported that the simultaneous addition of Sn and Ca can strongly refine the grains. However, in this paper, it can be seen from the figures that the un-DRXed grains in ZMXE3101 exhibit a blocky morphology, while the un-DRXed grains of ZTXE3101 display an elongated and strip-like appearance. This observation suggests that the Ca-MM(La-Ce) combination has a more prominent effect on the high-temperature softening behavior of the ZT31 alloy. Furthermore, elongated strip-like un-DRXed grains contribute to enhancing the strength of the alloy in the extrusion direction.

Figure 3h shows the dislocation distribution in equiaxed grains, indicating that the deformation mechanism of equiaxed grains during deformation is primarily dislocation slip mechanism dominated. As observed in Figure 3j, the deformation bands were found in the ZTXE3101 alloys with a width less than 1 μm, typically occurring in un-DRXed grains. These deformation bands are formed due to the interaction of oppositely signed edge dislocations within parallel slip planes, causing the crystal to bend. Such deformation bands generally require only a few microseconds to randomize the internal orientation of un-DRXed grains, which facilitates plastic deformation [24]. However, such deformation bands were not observed in the non-recrystallized structure of the ZM series alloys, which may explain the difficulty in achieving plastic deformation in the un-DRXed microstructure of the ZMXE3101 alloy. Further investigation is needed to understand why deformation bands are more readily formed in the un-DRXed microstructure of the ZMXE3101 alloy.

### 3.3. Mechanical Properties

The engineering stress–strain curves and true stress–strain curves of each specimen at room temperature are depicted in Figure 4a–d. Figure 4e shows the corresponding hardness values, with the values of 55.3, 69.5, 54.8, and 68.6, respectively. It reveals that the ZTXE3101 alloy exhibits superior mechanical properties, with the highest yield strength of 327.1 MPa and tensile strength of 328.3 MPa. However, it is noteworthy that this alloy also demonstrates the lowest ductility, measured at a mere 8.5% elongation. A comparative assessment of the ZM31 and ZT31 alloy samples indicates that the addition of Mn in Mg-Zn alloys is more effective in augmenting the strength. For the ZM31 alloy specifically, the composite addition of Ca-La-Ce elements not only enhances the strength but also exhibits a marginal improvement in ductility, evidenced by an increase from 18.2% to 18.7%. In contrast, while the ZT31 alloy system lags behind the ZM31 in terms of strength, the incorporation of Ca-La-Ce elements significantly boosts its strength, albeit with a substantial compromise in ductility, resulting in a near-halving of its plasticity.

The strain-hardening exponent actually reflects the strain homogenization ability of materials [25]. In order to further quantify the strain-hardening behavior of the as-extruded Mg-Sn alloys, the uniform plastic deformation stage in the uniaxial tensile curve is fitted by the Holloman equation [26]:
$$\sigma =K\varepsilon_{t}^{n}$$
where K is the strength coefficient and *n* is the strain-hardening exponent. The corresponding data are presented in Table 3.

The ZM31 and ZT31 alloys exhibit relatively higher strain-hardening exponents (*n*) as shown in Table 3, indicating their superior secondary formability. In contrast, while the ZTXE3101 alloy possesses higher strength, the ZMXE3101 alloy displays a higher n value, suggesting that the ZTXE3101 alloy holds greater practical engineering value as a structural material; however, ZMXE3101 has a better secondary processing performance.

Chaoyue Zhao previously reported a strain-hardening exponent of 0.21 for a Mg-1Sn alloy [23]. Notably, the strain-hardening exponent of Mg-3Zn-1Sn in this study is nearly twice that of Mg-1Sn, indicating that the addition of Zn to Mg alloys significantly enhances their secondary formability. Numerous studies have demonstrated that the simultaneous addition of Ca and rare-earth elements can enhance the strength of magnesium alloys to 350 MPa [26,27]. However, the alloy in this study exhibits superior properties and secondary formability, which are seldom mentioned in other research.

### 3.4. Texture Evolution

Figure 5 presents the EBSD images and recrystallized grain size distributions (derived from EBSD data analysis). From Figure 6a–d, it can be observed that the relationship of the recrystallized grain sizes among these four alloys is ZTXE3101 (1.3 um) < ZMXE3101 (2.0 um) < ZM31 (4.4 um) < ZT31 (6.6 um).

Given the strong extrusion texture exhibited by the deformed grains in magnesium alloys, recrystallized regions for the EBSD experiments were selected to be observed. The EBSD maps, pole Figures (PF) for {0001}, {10-10}, and {11-20} planes, and inverse pole Figures (IPF) in the ED direction are presented in Figure 6.

As evident from Figure 6a,b, there is little variation in the texture morphology between the ZM31 and ZMXE3101 alloys. Both exhibit a partial {0001} plane parallel to the ND direction, with {10-10} and {11-20} planes parallel to the ED direction. However, the basal texture intensity of ZM31 is significantly higher than that of the ZMXE3101 alloy. Upon examining their inverse pole figures, ZM31 exhibits a rare-earth texture, generally recognized for significantly enhancing the ductility of Mg alloys [28,29]. Furthermore, the presence of partial extension twins in the ZM31 alloy are also beneficial for the ductility [30]. These extension twins contribute favorably to its strain-hardening exponent (n-value) [31].

In summary, although ZM31 exhibits a relatively strong basal texture morphology, its rare-earth texture component and extension twins contribute to its plastic deformation, thus ensuring good ductility. On the other hand, while the addition of Ca-La-Ce transforms the rare-earth texture into an extrusion texture, specifically <10-10>//ED, the weakened basal texture compensates for any reduction in ductility caused by this change. Therefore, there is no significant decrease in the elongation of the ZMXE3101 alloy.

As shown in Figure 5, the refinement effect of ZT31 on recrystallized grains is stronger than that of ZM31, and ZT31 also produces extension twins, which explains its highest strain-hardening exponent (n-value). Unlike the ZM series alloys, the addition of Ca-La-Ce elements has a significant impact on the texture morphology of the ZT series alloys. In ZT31, the {0001} plane is parallel to the ND direction, while in ZTXE3101, the {0001} plane is parallel to the TD direction, with a basal texture intensity reaching 13.38. Moreover, the addition of Ca-La-Ce elements significantly increases the texture intensity of <10-10>//ED and enhances the prismatic <10-10>//ED texture intensity from 4.07 to 18.91.

As the grain size decreases, the critical resolved shear stress (CRSS) for the prismatic slip also decreases; when the grain refinement reaches a certain level, the prismatic slip becomes one of the important deformation mechanisms for the plasticity of magnesium alloys [32,33,34]. Therefore, when the grains are relatively fine, the basal and prismatic texture morphologies of deformed magnesium alloys are both important. The intense basal and prismatic textures in ZTXE3101 are significant factors contributing to its reduced ductility.

Figure 6e depicts the EBSD map and corresponding pole Figures and inverse pole Figure of the recrystallized region of the ZTXE3101 alloy. It is evident from the Figure that the orientations of various faces of the DRXed grains in ZTXE3101 alloy are highly random. The prior literature has indicated that if the recrystallized structure possesses good ductility, it can coordinate the plastic deformation of un-DRXed grains, leading to excellent comprehensive mechanical properties [18,23]. As shown in Figure 6e, the DRXed region of the ZTXE3101 alloy exhibits a typical <10-10>//ED texture, which is neither conducive to the slip of the {0001} planes nor to the slip of the {10-10} planes. Therefore, it can be concluded that the plastic potential of the DRXed grains of the ZTXE3101 alloy is quite limited, and its overall low ductility is attributed to the aforementioned reasons.

It is worth noting that the as-extruded alloys with rare-earth elements exhibit hardly any observable twins, indicating that the presence of rare-earth elements weakens the twinning behavior. This is attributed to two aspects: firstly, the refined grain size favors suppressing the critical resolved shear stress (CRSS) of the twins; secondly, the grain refinement also reduces the CRSS for the prismatic and pyramidal slip. The generation of twins is closely related to the work-hardening rate, and an increase in twins leads to a higher work-hardening exponent, consistent with the results in mechanical properties. Deformed grains often produce twins during plastic deformation, but they often lack active dislocations, making them contribute primarily to strength while having minimal contribution to work hardening. This results in a weak work-hardening phenomenon in dual-peak grain structures [11,17,35].

Figure 7 shows the Schmid factor distribution of the (0002) <11-20> base slip for three extruded sheets. As can be seen from the graph, the relationship of the Schmid factor values among the four alloys is ZMXE3101 > ZM31 > ZT31 > ZTXE3101. The Schmid factor provides a good explanation for the plasticity relationship among these four alloys. Since the SF values are measured by the EBSD data from the recrystallized regions for most of the samples, as shown in Figure 6e, in the alloys, this suggests that the ductility of the Mg-3Zn-1Mn/Sn alloys primarily depends on the orientation distribution of the DRXed grains rather than un-DRXed ones.

## 4. Conclusions

In this study, a comparative investigation was conducted to analyze the influence of Ca-La-Ce on the mechanical properties and microstructural evolution of Mg-Zn-Mn/Sn alloys. The following conclusions were drawn from this research:(1)The mechanical strengths observed in the Mg-3Zn-1Mn alloy are higher than those of the Mg-3Zn-1Sn alloy, which can be attributed to the pronounced restraining effect exerted by nanosized Mn particles on the recrystallization and grain growth processes, surpassing the constraining capability of Mg_2_Sn phases.(2)The addition of Ca-La-Ce elements significantly improves the strength of the Mg-3Zn-1Sn alloy, resulting in an approximately 92.5% increase in the yield strength (YS), an approximately 29.2% enhancement in the ultimate tensile strength (UTS), and a nearly 52.2% reduction in the elongation (EL).(3)The effect of Ca-La-Ce on recrystallization and grain refinement in the Mg-3Zn-1Sn alloys is significantly greater than that in the Mg-3Zn-1Mn alloys, and the ZTXE3101 alloy exhibits a tendency to form fibrous non-recrystallized structures, which subsequently enhance its strength. In contrast, the ZMXE3101 alloy possesses excellent comprehensive mechanical properties due to its optimized texture distribution. The optimized texture distribution in the ZMXE3101 alloy is a key factor that contributes to its balanced and superior mechanical performance.

This study preliminarily investigates the influence patterns of light rare earths, Mn, Sn, and Ca on the microstructural evolution and mechanical properties of wrought magnesium alloys based on Mg-3Zn alloys. Although this kind of Mg alloy may not achieve the high strength comparable to highly alloyed Mg alloys or Mg-Mn-Ca-Sn alloys, as far as the authors’ limited practical experience, the Mg alloys of this system exhibit superior processability, making them more suitable for large-scale production. Naturally, there remains a substantial amount of work to be performed in the future, such as optimizing the alloy compositions and investigating the effects of subsequent heat treatments on the properties.

## Figures and Tables

**Figure 1 materials-17-04840-f001:**
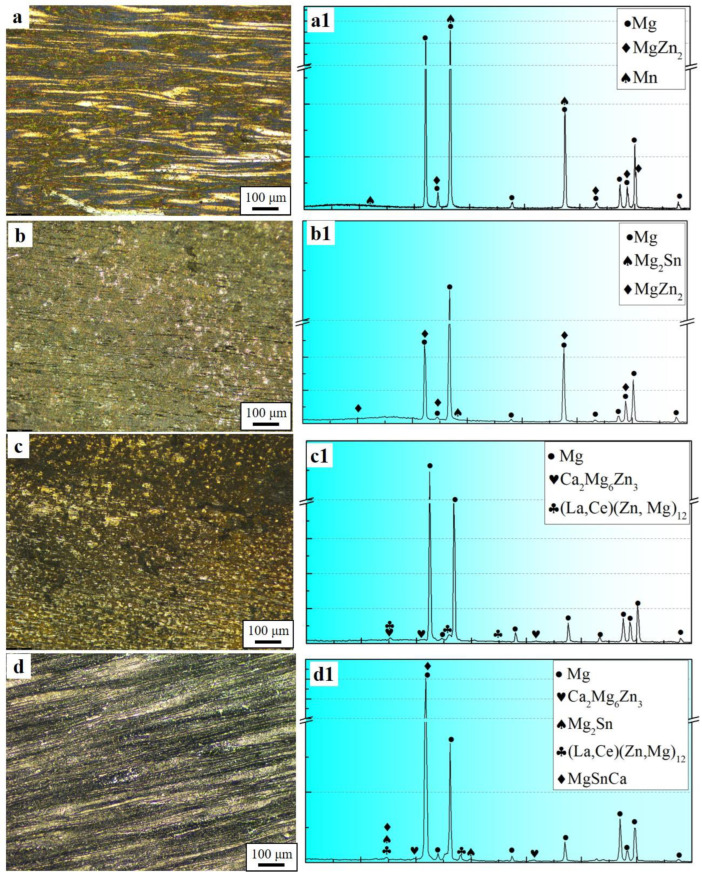
Optical micrographs and XRD diffraction patterns of the extruded rods: (**a**,**a1**) ZM31, (**b**,**b1**) ZT31, (**c**,**c1**) ZMXE3101, (**d**,**d1**) ZTXE3101.

**Figure 2 materials-17-04840-f002:**
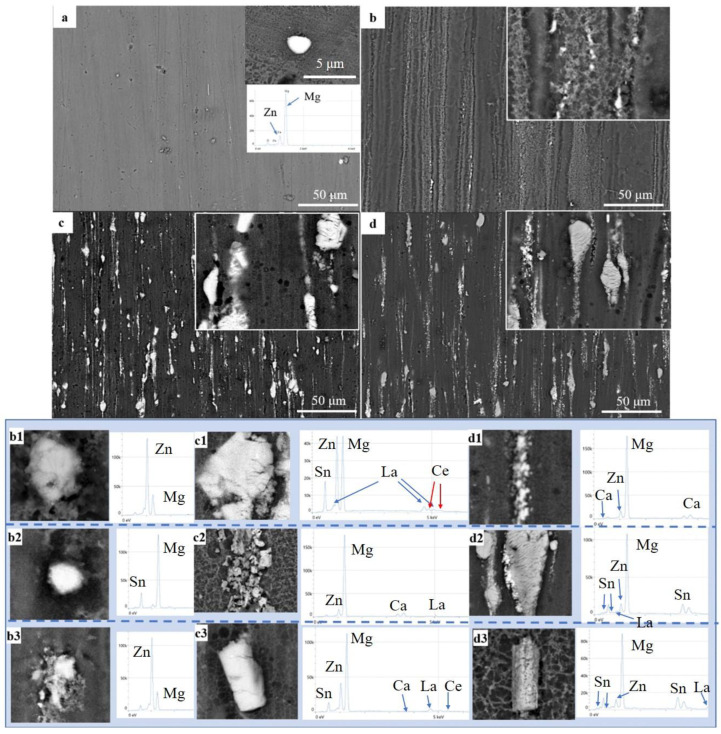
Low-magnification SEM morphologies show the distribution of second phases: (**a**) ZM31, (**b**) ZT31, (**c**) ZMXE3101, (**d**) ZTXE3101; and the typical second-phase morphologies with corresponding EDS analysis results: (**b1**–**b3**) ZT31, (**c1**–**c3**) ZMXE3101, (**d1**–**d3**) ZTXE3101.

**Figure 3 materials-17-04840-f003:**
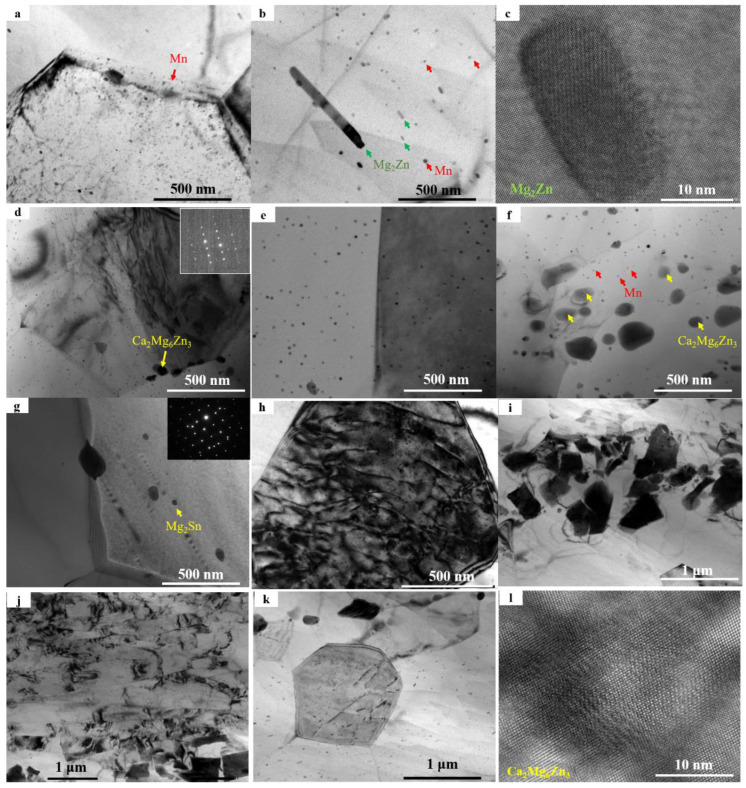
Low- and high-magnification TEM images of as-extruded alloys: (**a**–**c**) ZM31, (**d**–**f**) ZMXE3101, (**g**) ZT31, (**h**–**l**) ZTXE3101 alloys.

**Figure 4 materials-17-04840-f004:**
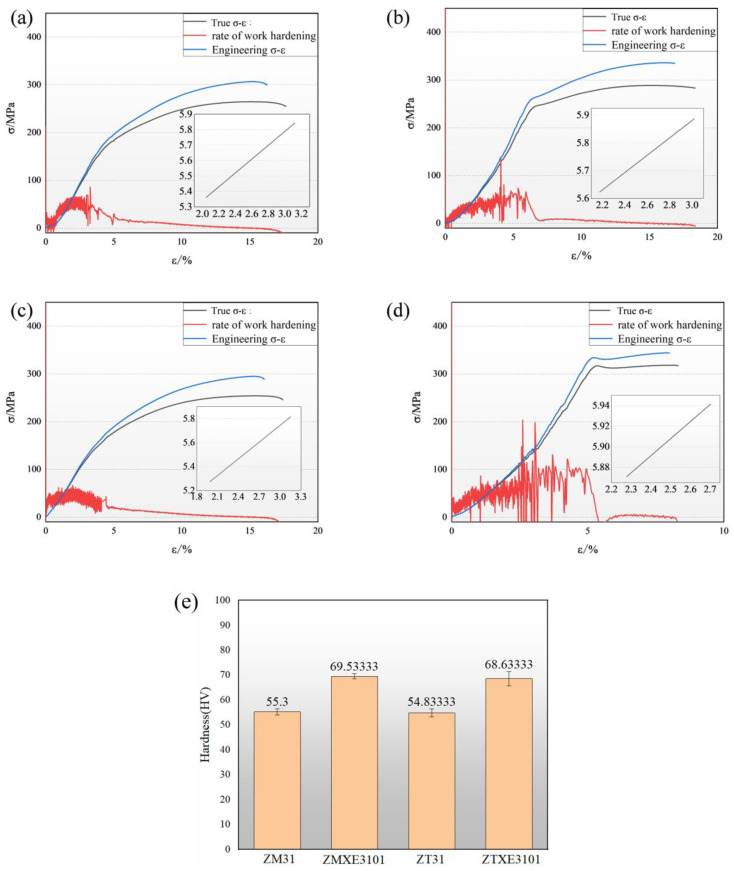
Engineering stress vs. strain curves and hardness values of the present as-extruded samples: (**a**) ZM31, (**b**) ZMXE3101, (**c**) ZT31, (**d**) ZTXE3101, (**e**) hardness values.

**Figure 5 materials-17-04840-f005:**
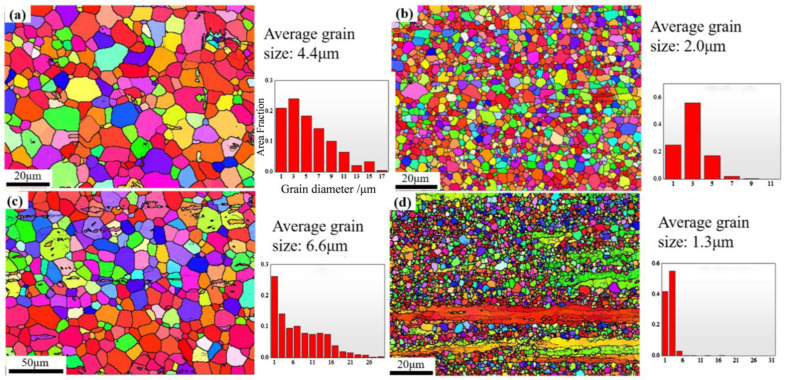
EBSD images and grain size distribution: (**a**) ZM31, (**b**) ZMXE3101, (**c**) ZT31, (**d**) ZTXE3101.

**Figure 6 materials-17-04840-f006:**
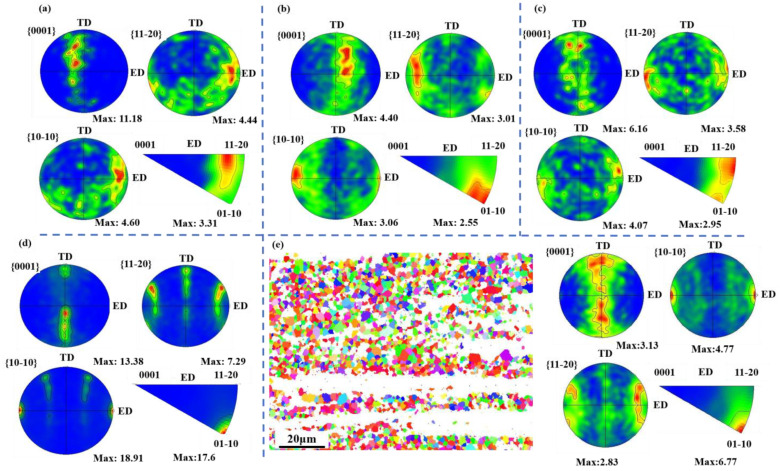
Pole Figures of {0001} {10-10} and {11-20} crystal planes, inverse pole Figures of ED, inverse pole Figures of ED: (**a**) ZM31, (**b**) ZMXE3101, (**c**) ZT31, (**d**) ZTXE3101. The pole Figure and inverse pole Figure of the recrystallized microstructure in ZTXE3101 alloy: (**e**).

**Figure 7 materials-17-04840-f007:**
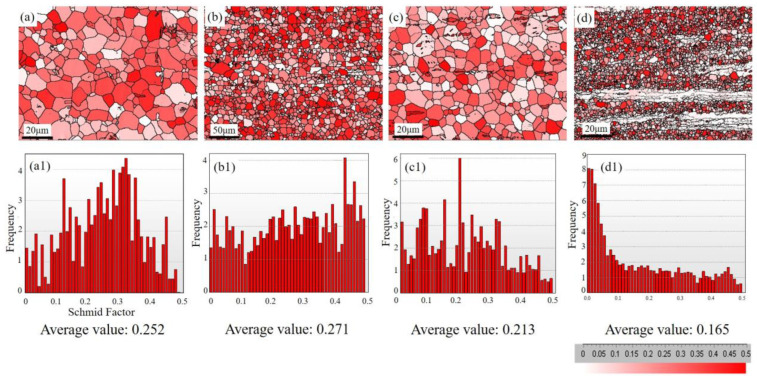
Basal Schmid factor distributions measured by EBSD for as-extruded alloys: (**a**,**a1**) ZM31; (**b**,**b1**) ZMXE3101; (**c**,**c1**) ZT31; (**d**,**d1**) ZTXE3101.

**Table 1 materials-17-04840-t001:** Chemical composition of the alloy determined by ICP-AES.

Alloy	Nominal Composition (wt.%)	Actual Composition (wt.%)						
		Mg	Zn	Ca	Mn	Sn	La	Ce
ZM31	Mg-3Zn-1Mn	Balance	2.91	0.45	1.21	-	1.32	0.66
ZMXE3101	Mg-3Zn-1Mn-0.5Ca-1La-0.5Ce	Balance	2.89	0.51	0.93	-	1.26	0.63
ZT31	Mg-3Zn-1Sn	Balance	2.68	0.42	-	0.87	1.02	0.51
ZTXE3101	Mg-3Zn-1Sn-0.5Ca-1La-0.5Ce	Balance	3.20	0.47	-	1.15	1.10	0.55

**Table 2 materials-17-04840-t002:** EDS results in Figure 2.

	Mg	Zn	Ca	Mn	Sn	La	Ce
b1	69.8	4.7	-	-	0.4	-	-
b2	84.9	-	-	-	13.5	-	-
b3	52.4	38.5	-		9.2	-	-
c1	64.7	30.5	-	-	-	3.1	1.8
c2	81.0	14.8	-	-	-	2.7	1.1
c3	59.6	10.8	0.3	-	-	2.0	0.8
d1	90.5	6.9	1.9	-	-	0.6	-
d2	61.2	32.4	-	-	-	4.3	1.7
d3	78.4	4.5	-	-	12.4	2.7	1.9

**Table 3 materials-17-04840-t003:** Mechanical properties of the alloys at room temperature.

Alloys	Tension Test				
	YS	UTS	δT	K	*n*
	(MPa)	(MPa)	(%)		
ZM31	180.9	264.6	18.2	4.44	0.45
ZMXE3101	246.8	288.7	18.7	4.94	0.31
ZT31	169.8	254.1	17.8	4.35	0.47
ZTXE3101	327.1	328.3	8.5	5.49	0.17

## Data Availability

The original contributions presented in the study are included in the article, further inquiries can be directed to the corresponding author.

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
