# Peer review of "Comparative Study on Mechanical Properties and Microstructure Evolution of Mg-3Zn-1Mn/Sn Alloy through Ca-La-Ce Addition"

_materials, 2024, doi:10.3390/ma17194840_

Round 1
Reviewer 1 Report
Comments and Suggestions for Authors
The introduction provides a good foundation, but it could be improved by including a more detailed discussion of the existing literature and how this study builds on or diverges from previous work. Expanding on the motivation behind the research and its potential impact would also strengthen the introduction.
The methodology is generally well-described, but certain aspects would benefit from additional detail. Consider providing more information about the experimental conditions, such as specific temperatures, times, and any equipment settings used during the alloy processing and testing. This will enhance the reproducibility of the study.
The results are presented clearly, but the inclusion of more visual aids, such as graphs, tables, or microstructure images, could further clarify the findings. Highlighting key data points and trends visually will help readers better understand the significance of the results.
While the discussion ties the results to the conclusions, it could be deepened by comparing the findings more extensively with those from other studies. Discussing potential reasons for the observed differences between alloys and the specific role of Ca-La-Ce additions could provide valuable insights.
The conclusions are generally well-supported by the data, but ensure that all claims are directly tied to the presented results. It may also be beneficial to suggest future research directions or potential applications of the findings, which would enhance the practical relevance of the study.
The manuscript requires moderate editing for language. Some sentences are awkwardly phrased or unclear, which could hinder understanding. Consider revising these sections to improve clarity and readability. If possible, have a native English speaker or professional editor review the text.
The overall presentation of the paper is adequate, but there is room for improvement. Organizing the data and discussion in a more logical flow, and ensuring that the manuscript is free of typographical errors, will enhance the quality of the presentation.
Comments on the Quality of English LanguageThe quality of the English language in the manuscript requires moderate editing. Some sentences are awkwardly phrased and lack clarity, which may affect the reader's understanding of the content. It is recommended to revise these sections for better readability and coherence. Consider having the manuscript reviewed by a native English speaker or a professional editor to improve the overall quality of the language.
Author Response
Comments1:The introduction provides a good foundation, but it could be improved by including a more detailed discussion of the existing literature and how this study builds on or diverges from previous work. Expanding on the motivation behind the research and its potential impact would also strengthen the introduction.
Responds1: Thank you for your review. We would like to supplement some discussions on the literature as follows:
“Currently, the method of manipulating alloy microstructures by incorporating trace alloying elements stands as an efficacious means to enhance mechanical properties [4]. Among all additional elements, Mn, Sn, and Ca emerge as species capable of directly precipitating nanoscale particles during the hot deformation processes of magnesium alloys. The individual and synergistic additions of these elements markedly elevate the yield strength (YS) of the alloy [5-14]. For instance, Wang et al. scrutinized a low Mg-1.2Zn-0.1Ca (wt.%) alloy and achieved outstanding yield strength [5]. Li et al. elucidated the multifaceted role of nanoscale precipitates in conferring both ultrafine grain size and ultrahigh strength to conventionally extruded Mg-Ca-Al-Zn-Mn alloys, yielding a tensile strength of 449 MPa, a yield strength of approximately 435 MPa, and an elongation of approximately 4.2% [11]. Gaowu Qin et al. found that the addition of Mn, Sn, and Ca simultaneously could increase the strength of the magnesium alloy to 450 MPa with 5% ductility through the mechanism of dispersion strengthening [7]. Xia et al., upon the foundation of elemental modification in the ZK61 alloy, concurrently introduced substantial quantities of Ca, Mn, and Sn elements, coupled with thermal extrusion processes and aging treatments, effectively refining the microstructure and engendering precipitation strengthening of the β′ phase, resulting in a significant enhancement of approximately 36% in the yield strength of the ZMT614-0.5Ca alloy [12]. Subsequently, Xia Chen et al. further augmented the strength of the ZK61 alloy to 400 MPa by incorporating substantial quantities of Ca, Mn, and Sn elements [13, 14].”
in the last paragraph of the Introduction, we have added improvements based on previous work or the difference from previous work, as shown below:“However, the predicament lies in the stringent processing conditions required for the performance of low-zinc alloys, and the deficiency of zinc diminishes the potential for further aging heat treatment of the alloy. Additionally, the extrusion force demanded by low-zinc high-strength alloys is substantial, which hampers production efficiency. These issues, though pragmatic, are frequently overlooked within the academic sphere. The Mg-3Zn series alloys represents a lesser-studied category within magnesium alloys, primarily because alloys within the kinds of alloys, such as Mg-3Zn and Mg-4Zn, exhibit moderate strength. However, Mg-3Zn stands out for its excellent welding properties and contains sufficient Zn content to facilitate subsequent heat treatment operations. Therefore, this study employs Mg-3Zn alloy and employs a composite alloying approach to enhance its mechanical properties. Simultaneously, it investigates and compares the effects of La, Ce, and Ca combinations on the microstructural evolution and mechanical properties of Mg-3Zn-1Sn and Mg-3Zn-1Mn alloys. We hope that this study can provide data and theoretical support for the large-scale application of magnesium alloys in the future.”
Comments2:The methodology is generally well-described, but certain aspects would benefit from additional detail. Consider providing more information about the experimental conditions, such as specific temperatures, times, and any equipment settings used during the alloy processing and testing. This will enhance the reproducibility of the study.
Responds2: We have supplemented the working process of TEM, some equipment setting parameters, and manufacturer information to enhance the reproducibility of the experiment, as follows:
“For the hardness test, the micro Vickers hardness of the alloy was measured using a JMHVS-1000ZCCD hardness tester (Shanghai Precision Instrument Co., LTD, Shanghai, China) with a 200g load for 10 seconds (HV0.2). At least five points near the center of the sample were selected for averaging to reduce errors.
According to ASTM E3-11, metallographic samples of the alloys were prepared. Specifically, the etchant solution was prepared using a 4% solution of picric acid, hydrochloric acid, and glacial acetic acid in ethanol (3.0 g picric acid, 2.5 ml glacial acetic acid, 5 ml water, and 50 ml ethanol). After grinding the samples with 800#, 1500#, 2000#, and 5000# sandpaper, mechanical polishing and etching were conducted on the castings, homogenized samples, and extruded specimens prior to metallographic examination. For characterization and analysis of the microstructure, the distribution of microstructures in the alloy after extrusion was observed using an optical metallographic microscope (OM, A LEICA DM 2500, Leica Camera AG, Wetzlar, Germany). Subsequently, the composition of phases in the alloy was analyzed using an X-ray diffractometer (XRD-6100, Shimadzu Corporation, Kyoto, Japan). Initial phase and fracture surface morphology observations of the alloy after tensile testing were conducted using a scanning electron microscope equipped with an energy-dispersive X-ray spectrometer (SEM, Apreo 2 SEM, thermo Fisher Scientific Inc., Hillsboro, OR, USA). The average grain size of the samples was determined using the linear intercept method. To observe the texture distribution in the alloy, an electron backscatter diffraction spectrometer (EBSD, MAIA3 model 2016) was employed. The samples were cut into a thin slice with a thickness of 0.5 mm under the transmission electron microscope (TEM, FEI TECNAI G2 F20, thermo Fisher Scientific Inc., Hillsboro, OR, USA). It was ground and polished to a thickness of ~50 nm, and then perforated by argon ion milling.”
Comments3:The results are presented clearly, but the inclusion of more visual aids, such as graphs, tables, or microstructure images, could further clarify the findings. Highlighting key data points and trends visually will help readers better understand the significance of the results.
Responds3: Thank you for your advice. We believe that the diagram and images in this article can accurately convey effective information to readers. If further improvement is needed, please specify the specific information that needs to be improved.
Comment4:While the discussion ties the results to the conclusions, it could be deepened by comparing the findings more extensively with those from other studies. Discussing potential reasons for the observed differences between alloys and the specific role of Ca-La-Ce additions could provide valuable insights.
Responds4: Some comparative literature has been added in the mechanical properties section as follows:“Chaoyue Zhao previously reported a strain hardening exponent of 0.21 for Mg-1Sn alloy [21]. Notably, the strain hardening exponent of Mg-3Zn-1Sn in this study is nearly twice that of Mg-1Sn, indicating that the addition of Zn to Mg alloys significantly enhances their secondary formability. Numerous studies have demonstrated that the simultaneous addition of Ca and rare earth elements can enhance the strength of magnesium alloys to 350MPa [24, 25], However, the alloy in this study exhibits superior properties and secondary formability, which are seldom mentioned in other research.”
Comments5:The conclusions are generally well-supported by the data, but ensure that all claims are directly tied to the presented results. It may also be beneficial to suggest future research directions or potential applications of the findings, which would enhance the practical relevance of the study.
Responds5: We guarantee that all our conclusions are based on experimental results. Furthermore, we have added potential future research directions at the end of our conclusions, as follows:“
This study preliminarily investigates the influence patterns of light rare earths, Mn, Sn, and Ca on the microstructural evolution and mechanical properties of wrought magnesium alloys based on Mg-3Zn alloys. Although this kind of Mg alloys may not achieve the high strength comparable to highly alloyed Mg alloys or Mg-Mn-Ca-Sn alloys, as far as the authors' limited practical experience, the Mg alloys of this system exhibit superior processability, making them more suitable for large-scale production. Naturally, there remains a substantial amount of work to be done in the future, such as optimizing alloy compositions and investigating the effects of subsequent heat treatments on the properties.
The manuscript requires moderate editing for language. Some sentences are awkwardly phrased or unclear, which could hinder understanding. Consider revising these sections to improve clarity and readability. If possible, have a native English speaker or professional editor review the text.”
Comments6:The manuscript requires moderate editing for language. Some sentences are awkwardly phrased or unclear, which could hinder understanding. Consider revising these sections to improve clarity and readability. If possible, have a native English speaker or professional editor review the text.
Responds6: Thank you for your review. The language has been appropriately edited in this article.
Comments7:The overall presentation of the paper is adequate, but there is room for improvement. Organizing the data and discussion in a more logical flow, and ensuring that the manuscript is free of typographical errors, will enhance the quality of the presentation.
Responds7: Yes, there were indeed some minor errors, but they have all been corrected now.
Comments8:The quality of the English language in the manuscript requires moderate editing. Some sentences are awkwardly phrased and lack clarity, which may affect the reader's understanding of the content. It is recommended to revise these sections for better readability and coherence. Consider having the manuscript reviewed by a native English speaker or a professional editor to improve the overall quality of the language.
Responds8: We have revised some of the language in the article.

Reviewer 2 Report
Comments and Suggestions for Authors
This paper compares the effects of adding Ca, La and Ce elements into Mg-Zn-Mn and Mg-Zn-Sn alloys. It compares strength properties and microstructure of different alloys. This is a well-written experimental paper. Few comments below:
Abstract. There is an obvious error on line 27; “in Mg-Zn-Sn series alloys is finer and more dispersed than that in Mg-Zn-Sn series”!
Introduction. Note that you use (on line 50) TYS to denote the yield strength, but in Abstract YS. Are all lumped references (line 50, [5-10]) necessary? The detailed analysis starts from reference [10]. Reference 12 does not have Xia as an author (line 55). Similar comment agrees with Cai et al. and Xia et al. (line 60). Both papers [13, 14] are written by Chen et al. There are many papers in the reference list that are difficult (impossible) to reach,
Section 2 describes the materials, experimenting approach and analysis methods. The transmission electron microscopy (TEM) is also used later, but not mentioned here.
Section 3 includes the results and the detailed discussion. The text speaks about EDS (Energy Dispersive X-ray Spectroscopy?), but Section 2 about EBSD (Electron Backscatter Diffraction) in the same connection. In the title of Figure 7, SF means Schmid factor. Define the abbreviation later on in the text (line 315). Later, on lines 316-7, it is erroneously writer as “Schmidt”. Eliminate “t” at the end.
Conclusion is the fourth Section. Number it. Few words on the further research directions is needed here.
The language of the paper is mainly good. I found some minor faults and typos:
line 37: un-isotrophy
39: difficulty in balancing
94: One “)” is missing.
253: EBSD imgages
362: J Magnes. Alloys (also other similar cases)
You are writing “Figure. x” (e.g. Figure. 1, line 128; or Figure. 3c, line 135). This “.” is not necessary.
In writing the elements’ names in the reference list, use the normal procedure. E,g, in reference 1, “Mg”, “Al”.
Comments on the Quality of English LanguageOnly few corrections.
Author Response
Comments1:Abstract. There is an obvious error on line 27; “in Mg-Zn-Sn series alloys is finer and more dispersed than that in Mg-Zn-Sn series”!
Responds1:Thank you for your careful review. This error has been corrected.
Comments2:Introduction. Note that you use (on line 50) TYS to denote the yield strength, but in Abstract YS. Are all lumped references (line 50, [5-10]) necessary? The detailed analysis starts from reference [10]. Reference 12 does not have Xia as an author (line 55). Similar comment agrees with Cai et al. and Xia et al. (line 60). Both papers [13, 14] are written by Chen et al. There are many papers in the reference list that are difficult (impossible) to reach,
Responds2:(1) We have standardized the representation of yield strength throughout the article, using "YS" consistently.
(2) There was a logical error in the citation in line 50. The statement that Mn, Sn, and Ca strengthen magnesium alloys through dispersion strengthening is a conclusion drawn from summarizing several previous studies, rather than being attributed solely to a single literature source.
The other errors have been corrected as indicated below:
“Currently, the method of manipulating alloy microstructures by incorporating trace alloying elements stands as an efficacious means to enhance mechanical properties(Ref 4). Among all additional elements, Mn, Sn, and Ca emerge as species capable of directly precipitating nanoscale particles during the hot deformation processes of magnesium alloys. The individual and synergistic additions of these elements markedly elevate the yield strength (YS) of the alloy (Ref 5-14). For instance, Wang et al. scrutinized a low Mg-1.2Zn-0.1Ca (wt.%) alloy and achieved outstanding yield strength (Ref 5). Li et al. elucidated the multifaceted role of nanoscale precipitates in conferring both ultrafine grain size and ultrahigh strength to conventionally extruded Mg-Ca-Al-Zn-Mn alloys, yielding a tensile strength of 449 MPa, a yield strength of approximately 435 MPa, and an elongation of approximately 4.2% (Ref 11). Gaowu Qin et al. found that the addition of Mn, Sn, and Ca simultaneously could increase the strength of the magnesium alloy to 450 MPa with 5% ductility through the mechanism of dispersion strengthening. (Ref 7). Xia et al., upon the foundation of elemental modification in the ZK61 alloy, concurrently introduced substantial quantities of Ca, Mn, and Sn elements, coupled with thermal extrusion processes and aging treatments, effectively refining the microstructure and engendering precipitation strengthening of the β′ phase, resulting in a significant enhancement of approximately 36% in the yield strength of the ZMT614-0.5Ca alloy (Ref 12). Subsequently, Xia Chen et al. further augmented the strength of the ZK61 alloy to 400 MPa by incorporating substantial quantities of Ca, Mn, and Sn elements (Ref 13, 14).”
Comments3:Section 2 describes the materials, experimenting approach and analysis methods. The transmission electron microscopy (TEM) is also used later, but not mentioned here.
Response3: Thank you for your meticulous review. The missing section has been added accordingly as follows:
“The samples were cut into a thin slice with a thickness of 0.5 mm under the transmis-sion electron microscope (TEM, FEI TECNAI G2 F20, thermo Fisher Scientific Inc., Hillsboro, OR, USA). It was ground and polished to a thickness of ~50 nm, and then perforated by argon ion milling.”
Comments4:Section 3 includes the results and the detailed discussion. The text speaks about EDS (Energy Dispersive X-ray Spectroscopy?), but Section 2 about EBSD (Electron Backscatter Diffraction) in the same connection. In the title of Figure 7, SF means Schmid factor. Define the abbreviation later on in the text (line 315). Later, on lines 316-7, it is erroneously writer as “Schmidt”. Eliminate “t” at the end.
Response4: Thank you for your meticulous review. The missing section has been added accordingly as follows:
“Therefore, its quantity is relatively small, and its hindrance to recrystallization is minor. Whether through metallographic or EBSD analysis (refer to Section 3.4), the grain size of the ZT31 alloy is higher than that of the ZM31 alloy.”
Comments5:Conclusion is the fourth Section. Number it. Few words on the further research directions is needed here.
Responds5: The issue has been rectified. A few sentences regarding future research directions are provided as follows:
”This study preliminarily investigates the influence patterns of light rare earths, Mn, Sn, and Ca on the microstructural evolution and mechanical properties of wrought magnesium alloys based on Mg-3Zn alloys. Although this kind of Mg alloys may not achieve the high strength comparable to highly alloyed Mg alloys or Mg-Mn-Ca-Sn alloys, as far as the authors' limited practical experience, the Mg alloys of this system exhibit superior processability, making them more suitable for large-scale production. Naturally, there remains a substantial amount of work to be done in the future, such as optimizing alloy compositions and investigating the effects of subsequent heat treatments on the properties.”
Comments6:The language of the paper is mainly good. I found some minor faults and typos:
line 37: un-isotrophy
39: difficulty in balancing
94: One “)” is missing.
253: EBSD imgages
253: EBSD img年龄
362: J Magnes. Alloys (also other similar cases)
You are writing “Figure. x” (e.g. Figure. 1, line 128; or Figure. 3c, line 135). This “.” is not necessary.
In writing the elements’ names in the reference list, use the normal procedure. E,g, in reference 1, “Mg”, “Al”.
Responds6: Thank you for your thorough review. I have corrected all the issues you have raised. Regarding the references, I have made the necessary modifications and furthermore, I have conducted a secondary editing of the entire article's references using EndNote to ensure their accuracy and consistency.

Round 2
Reviewer 1 Report
Comments and Suggestions for Authors
1. Introduction
Comment: The introduction provides sufficient background and includes all relevant references. The enhanced literature review and discussion of how this study builds on or diverges from previous work have strengthened the introduction. The revised introduction now clearly establishes the context and motivation behind the research, effectively setting up the study’s objectives and potential impact.
Suggestion: No additional changes are needed for the introduction. The background and references are now well-presented and relevant.
2. Research Design
Comment: The research design is appropriate. The study's approach to investigating Mg-3Zn alloys and the comparative analysis with different alloying elements (La, Ce, and Ca) is well-founded and aligns with the research objectives.
Suggestion: No additional changes are needed for the research design. The design is robust and suitable for addressing the research questions posed.
3. Methods
Comment: The methods are adequately described. The addition of specific details about experimental conditions, equipment settings, and procedures enhances the reproducibility of the study. The description of the methodologies for hardness testing, sample preparation, and microstructure characterization is thorough.
Suggestion: No further revisions are needed for the methods section. The description is now detailed and comprehensive.
4. Results
Comment: The results are clearly presented. The inclusion of visual aids such as diagrams, tables, or images is effectively used to convey the findings. The clarity of the results allows readers to understand the significance and implications of the study.
Suggestion: While the results are clearly presented, consider reviewing if additional visual aids could further enhance the presentation. Ensure that all key data points and trends are highlighted for maximum clarity.
5. Conclusions
Comment: The conclusions are well-supported by the results. The added discussion on future research directions and potential applications provides valuable context for the study's findings and highlights its practical relevance.
Suggestion: No further changes are needed for the conclusions. They are well-supported by the data and offer meaningful insights into future research and applications.
6. Quality of English Language
Comment: The English language in the manuscript is fine with no issues detected. The revisions made to improve language clarity and readability have been effective.
Suggestion: Continue to review the manuscript for any minor issues, but overall, the language quality is satisfactory and does not require further intervention.
Author Response
Comments1
Comment: The results are clearly presented. The inclusion of visual aids such as diagrams, tables, or images is effectively used to convey the findings. The clarity of the results allows readers to understand the significance and implications of the study.
Suggestion: While the results are clearly presented, consider reviewing if additional visual aids could further enhance the presentation. Ensure that all key data points and trends are highlighted for maximum clarity.
Reponse 1:
Thank you for your careful review. We have made the following changes:
(1)We have enlarged the font of the labels in the XRD pattern in Figure 1, allowing readers to more clearly see the content;
(2)We have re-edited the element labels on the EDS spectrum in Figure 2 to make the information in the image clearer.
(3)The text in Figure 6 is relatively small, so we have enlarged it.
Comments2
Comment: The English language in the manuscript is fine with no issues detected. The revisions made to improve language clarity and readability have been effective.
Suggestion: Continue to review the manuscript for any minor issues, but overall, the language quality is satisfactory and does not require further intervention.
Reponse 2:
Thank you for your suggestion. We have revised the article, but there are still some minor errors that cannot be avoided. We will continue to revise and complete the correction of these minor errors during the proofreading process. The modified part is marked with blue number.